# Knowledge, attitude, and practice of the National Guidelines for Diagnosis and Treatment of Malaria among medical doctors in Ebonyi state, Nigeria: A cross-sectional survey

Ugwu I. Omale👤 *

Department of Community Medicine, Alex Ekwueme Federal University Teaching Hospital, Abakaliki (AEFUTHA), Ebonyi State, Nigeria

* omaleiu@gmail.com, omaleui@yahoo.com

## Abstract

### Background

The Guidelines for Diagnosis and Treatment of Malaria are meant to guide medical practitioners to enhance optimal management of patients suspected of having malaria which is vital to malarial control and elimination. Medical doctors have the main responsibility for treating patients with malaria so there was need to evaluate the extent to which the medical doctors in Ebonyi state, Nigeria, knew, viewed, and practiced the 2015 National Guidelines for Diagnosis and Treatment of Malaria.

### Methods

A cross-sectional survey was carried out in May, 2019 among medical doctors who were involved in the management of malaria patients and selected via convenience sampling. Data was collected using a structured self-administered questionnaire. Each respondent was categorized as having poor, fair, or good knowledge, attitude, and practice respectively based on knowledge, attitude, and practice scores of <50%, 50–75%, and >75%. Associated factors were evaluated through bivariate and multivariate binomial logistic regressions at 5% probability of type one error and the overall test for the effect of each independent factor on practice level was done using the "postestimation test" command in Stata.

### Results

A total of 156 medical doctors were surveyed. Most, 138 (88.5%) were aware there was a national guideline for diagnosis and treatment of malaria. Among the medical doctors, 19 (12.2%) had good knowledge, 110 (70.5%) had fair knowledge, and 27 (17.3%) had poor knowledge; 38 (24.4%) had good attitude, 74 (47.4%) had fair attitude, and 44 (28.2%) had poor attitude; and 59 (37.8%) had good practice, 90 (57.7%) had fair practice, and 7 (4.5%) had poor practice. The attitude level of the medical doctors toward the 2015 National

**Funding:** The author received no specific funding for this work.

**Competing interests:** The author has declared that no competing interests exist.

Guidelines for Diagnosis and Treatment of Malaria was a predictor of good practice (adjusted p-value of overall effect = 0.0003).

## Conclusions

Although most of the medical doctors in Ebonyi state, Nigeria, were aware of the existence of a national guideline, only just over a third of them had good practice of the 2015 National Guidelines for Diagnosis and Treatment of Malaria. Policy interventions should focus on improving the attitude of the medical doctors toward the National Guidelines through training and re-training.

## Introduction

Malaria is an endemic disease of public health importance in Nigeria and other high burden countries of the World Health Organization (WHO) Africa Region [1–3]. The Guidelines for Diagnosis and Treatment of Malaria are meant to guide medical practitioners to enhance optimal management of patients suspected of having malaria. Optimal management of patients suspected of having malaria, including prompt and accurate diagnosis (via parasitological diagnostic testing) and targeted treatment with effective anti-malarial drugs such as artemisinin-based combination therapies (ACTs), is vital to malarial control and elimination [3–6].

The 2015 Nigeria National Guidelines for Diagnosis and Treatment of Malaria was the third edition of such guidelines. There were several recommendations with regards to the diagnosis and treatment of malaria (both uncomplicated and severe malaria) in children, adults, and in pregnancy. Parasitological confirmation of malaria with microscopy or malaria rapid diagnostic test (MRDT) in all patients suspected of having malaria before treatment and restriction of treatment with anti-malarial drugs to only patients with positive diagnostic test results were emphasized by the 2015 Guidelines [7] and re-emphasized by the revised or 2020 Guidelines [8] which is now the current and fourth edition. These recommendations are in consonance with the WHO recommendation of universal parasitological diagnostic testing before malaria treatment [5, 9]. This recommendation aims to prevent the sole use of presumptive diagnosis (based on clinical symptoms) which is associated with over-diagnosis and over-treatment of malaria with the resultant increase in the risk of drug resistance [5, 9]. However, for universal parasitological diagnostic testing (with microscopy or MRDT) to solve this problem, health workers (including medical doctors) have to respond appropriately to negative diagnostic test results by not prescribing anti-malarial drugs [10].

For the National Guidelines for Diagnosis and Treatment of Malaria to achieve its aim, medical doctors, who have the main responsibility for treating patients with malaria, have to have good knowledge of the guidelines, good attitude towards the guidelines, and above all, practice (or follow) the guidelines in the day-to-day management of their patients suspected of having malaria. Although medical doctors are highly skilled medical practitioners who are expected to uphold optimal patient management in their daily practice, it has been reported that many medical doctors (in two tertiary health facilities in Nigeria) usually make diagnosis of malaria based on the sole use of presumptive diagnosis [11]. However, there are limited studies on knowledge, attitude, and practice of National Guidelines for Diagnosis and Treatment of Malaria among medical doctors. Some studies have assessed knowledge and use of malaria diagnostic methods among medical doctors [11, 12] and healthcare workers [13] while some have assessed healthcare workers' adherence to the recommended means of malaria

diagnosis and treatment for uncomplicated malaria [14–17]. However, the extent to which medical doctors adhere to these National Guidelines for Diagnosis and Treatment of Malaria in their daily practice has not been rigorously evaluated particularly in Nigeria. Moreover, the above studies only focused on uncomplicated malaria. To my knowledge, no such study has been done in Ebonyi state, Nigeria. A good understanding of medical doctors' knowledge and attitude, and the extent to which they practice the many salient recommendations of the Guidelines, as well as the associated factors, will guide related interventions and relevant policy decisions regarding the optimal management of patients suspected of having malaria among the medical doctors in Ebonyi state, Nigeria. This will enhance malaria control.

The aim of this study was to evaluate the knowledge, attitude, and practice of the 2015 National Guidelines for Diagnosis and Treatment of Malaria, and associated factors, among medical doctors in Ebonyi state, Nigeria.

## Methods

### Study design and setting

This study was an analytical cross-sectional survey carried out in May, 2019 in Ebonyi state which is in the south-east geopolitical zone of Nigeria. Malaria is endemic in Ebonyi state with year round transmission. Ebonyi state's projected population for 2020 was 3,251,508 based on the 2006 national census figure and a growth rate of 2.8% [18]. There are 13 Local Government Areas (LGAs) in the state. As of 2020, there were 784 orthodox health care facilities in the state: 566 public facilities (2tertiary, 13 secondary and 551 primary health facilities) and 218 private facilities (25 secondary and 193 primary health facilities) [19].

The two tertiary health facilities were the Alex Ekwueme Federal University Teaching Hospital Abakaliki (AEFUTHA) and the National Obstetrics Fistula Centre (NOFIC). The 13 public secondary health facilities were the 13 general hospitals in each of the 13 LGAs of the state while the 25 private secondary health facilities comprised of private and missionary hospitals [19]. According to the record of the Ebonyi state chapter of the Nigerian Medical Association, there are about 670 medical doctors in Ebonyi state in 2021 and most of them work primarily at AEFUTHA which provides primary, secondary, and tertiary health care for the people.

### Study participants and data collection

Medical doctors who were involved in the management of malaria patients and gave verbal consent were included in the study. They comprised of doctors in community medicine, family medicine, internal medicine, obstetrics and gynaecology, and paediatics departments in AEFUTHA as well as those in general hospitals, missionary hospitals, private hospitals, and medical centres. Convenience sampling of those that were available and consented to participate, during a visit to their respective departmental clinical meetings and places of work, was used to select 156 medical doctors.

A pre-tested structured self-administered questionnaire was used to collect data on sociodemographic and background characteristics, knowledge, attitude, and practice of the 2015 National Guidelines for Diagnosis and Treatment of Malaria. Multi-choice objective questions with 23 variables were used to measure their awareness and knowledge of salient recommendations of the Guidelines. Their attitude toward the Guidelines was measured through the extent to which they agreed or disagreed with 12 salient recommendations of the Guidelines by using a five-level response option: strongly disagree, disagree, undecided, agree, and strongly agree. Their practice of the Guidelines was measured through the frequency with which they practiced 13 salient recommendations of the Guidelines by using a five-level response option: never, rarely, sometimes, often, and always. Completed questionnaires were

reviewed immediately after collection for missing data and inconsistencies and some were returned to the respective respondents for correction.

## Ethics statement

Ethical approval for the study was obtained from the University Research Ethics Committee (UREC) of Ebonyi State University (number EBSU/DRIC/UREC Vol.05/067). Informed oral consent was obtained from the respondents during the survey. The purpose and nature of the study including the voluntary nature of participation was communicated to participants. They were also assured of confidentiality that the information provided would be strictly protected from unauthorized persons.

## Statistical analysis

Data was entered using Microsoft Excel 2007 (Microsoft Inc., Redmond, Washington), verified, and analysed using Stata (SE) version 15.1 (Stata Corp, College Station, TX, USA). The data was summarized using frequencies and proportions or percentages and presented in tables and graphs. The overall knowledge level of each respondent was derived from the knowledge scores for each respondent. Each right answer was scored "one" and each wrong answer was scored "zero". A total knowledge score was computed for each respondent with a total attainable score of 23. The total score obtained was converted to percentages and then graded on a three level scale such that scores of < 50% was taken to be poor knowledge, 50–75% as fair knowledge, and >75% as good knowledge.

Also, the overall attitude level of each respondent was derived from the individual attitude scores. For each of the statements in the attitude section of the questionnaire, the response of each respondent was scored as follows: "strongly agree" was scored "five", "agree" was scored "4", "undecided" was scored "two", "disagree" was scored "0", and "strongly disagree" was scored "0". A total attitude score was computed for each respondent with a total attainable score of 60. The total score obtained was converted to percentages and then graded on a three level scale such that scores of < 50% was taken to be poor attitude, 50–75% as fair attitude, and >75% as good attitude.

Similarly, the overall practice level for each respondent was derived from the individual practice scores. For each of the practice items in the questionnaire, the response of each respondent was scored as follows: where "never" was taken as the best practice, "never" was scored "five", "rarely" was scored "4", "sometimes" was scored "two", "often" was scored "0", and "always" was scored "0"; where "always" was taken as the best practice, "always" was scored "five", "often" was scored "4", "sometimes" was scored "two", "rarely" was scored "0", and "never" was scored "0". A total practice score was computed for each respondent with a total attainable score of 65. The total score obtained was converted to percentages and then graded on a three level scale such that scores of < 50% was taken to be poor practice, 50–75% as fair practice, and >75% as good practice.

The practice level was dichotomized as good and fair or poor practice and its association with independent factors (background characteristics, knowledge level, and attitude level) was tested at 5% probability of type one error via bivariate and multivariate binomial logistic regressions (controlling for the other independent factors). Since the number of the independent factors was modest, all were simultaneously entered into the multivariate regression model to minimize bias. The overall test for the effect of each independent factor on practice level was done using the "postestimation test" command in Stata. Crude and adjusted odds ratios, 95% confidence intervals and p-values are reported.

## Results

### Sociodemographic and background characteristics

Among the 156 medical doctors, the majority were aged between 31–40 years, 83 (53.2%) followed by those aged more than 40 years, 47 (30.1%). Majority of them, 108 (69.2%) were males and 48 (30.8%) were females. The highest proportion of the respondents: were senior registrars, 67 (43.0%); had 6–10 years of working experience, 61 (39.1%); were currently practicing at the Federal University Teaching Hospital, 134 (85.9%); and were in Community Medicine Department in the teaching hospital, 34 (21.8%) (Table 1).

### Knowledge of the 2015 National Guidelines for Diagnosis and Treatment of Malaria

Among the 156 medical doctors (Table 2): 138 (88.5%) were aware there was a national guideline for diagnosis and treatment of malaria; 81 (51.9%) had seen the current (2015) guidelines; 66 (42.3%) had read the current (2015) guidelines; 92 (59.0%) knew all patients suspected of

**Table 1. Sociodemographic and background characteristics of respondents.** Total = 156.

| Characteristic | n | % |
| --- | --- | --- |
| **Age group** (in years) | | |
| ≤30 | 26 | 16.7 |
| 31–40 | 83 | 53.2 |
| >40 | 47 | 30.1 |
| **Sex** | | |
| Male | 108 | 69.2 |
| Female | 48 | 30.8 |
| **Professional rank** | | |
| House officer | 29 | 18.6 |
| Registrar | 20 | 12.8 |
| Senior registrar | 67 | 43.0 |
| Medical officer | 27 | 17.3 |
| Consultant | 13 | 8.3 |
| **Duration of practice** (in years) | | |
| ≤5 | 37 | 23.7 |
| 6–10 | 61 | 39.1 |
| 11–15 | 30 | 19.2 |
| >15 | 28 | 18.0 |
| **Current primary place of practice** | | |
| Secondary health facility | 22 | 14.1 |
| Private hospital | 2 | 1.3 |
| Missionary hospital | 11 | 7.1 |
| General hospital | 6 | 3.8 |
| Federal polytechnique medical centre | 3 | 1.9 |
| Federal university teaching hospital | 134 | 85.9 |
| Community medicine department | 34 | 21.8 |
| Family medicine department | 17 | 10.9 |
| Internal medicine department | 26 | 16.7 |
| Paediatrics department | 29 | 18.6 |
| Obstetrics and Gynaecology dept. | 28 | 18.0 |

**Table 2. Knowledge of respondents about the 2015 National Guidelines for Diagnosis and Treatment of Malaria.** Total = 156.

| Knowledge items | n | % |
|---|---|---|
| Were aware there was a national guideline for diagnosis and treatment of malaria | 138 | 88.5 |
| Had seen the current (2015) guidelines | 81 | 51.9 |
| Had read the current (2015) guidelines | 66 | 42.3 |
| Knew all patients suspected of having uncomplicated malaria should receive treatment based on confirmation of diagnosis with microscopy or MRDT | 92 | 59.0 |
| Knew the results of parasitological diagnosis of uncomplicated malaria should be available within two hours of patients presenting | 15 | 9.6 |
| Knew anti-malarial drugs should be limited to only test positive cases | 84 | 53.8 |
| Knew parallel testing with microscopy and MRDT was not recommended | 74 | 47.4 |
| Knew treatment based solely on clinical suspicion should only be considered in children aged below five years when microscopy or MRDT was not accessible | 111 | 71.1 |
| Knew the recommended anti-malarial drug for uncomplicated malaria was ACTs | 154 | 98.7 |
| Knew the ACT of first choice was artemether-lumefantrine | 149 | 95.5 |
| Knew the ACT of second choice was artesunate-amodiaquine | 119 | 76.3 |
| Knew ACTs were the recommended anti-malarial drugs for lactating mothers with uncomplicated malaria | 131 | 84.0 |
| Knew ACTs were the recommended anti-malarial drugs for uncomplicated malaria in second and third trimesters of pregnancy | 113 | 72.4 |
| Knew quinine plus clindamycin for 7 days was the first choice anti-malarial treatment for uncomplicated malaria in first trimester of pregnancy | 48 | 30.8 |
| Knew ACTs were the second choice anti-malarial treatment for uncomplicated malaria in first trimester of pregnancy | 66 | 42.3 |
| Knew all patients suspected of having severe malaria should have a parasitological diagnosis before treatment | 69 | 44.2 |
| Knew parenteral artesunate was the recommended (first choice) anti-malarial drug for severe malaria | 100 | 64.1 |
| Knew parenteral artemether and quinine were the alternate anti-malarial drugs for severe malaria | 19 | 12.2 |
| Knew that, once started, parenteral anti-malarial drugs for severe malaria should be given for at least 24 hours | 73 | 46.8 |
| Knew the patient should be given the full dose of an ACT after parenteral anti-malarial drug for severe malaria | 142 | 91.0 |
| Knew parenteral artesunate was the recommended anti-malarial drug for severe malaria in second and third trimesters of pregnancy | 65 | 41.7 |
| Knew parenteral artesunate was the recommended anti-malarial drug for severe malaria in first trimester of pregnancy | 53 | 34.0 |
| **Among those that were aware there was a national guideline for diagnosis and treatment of malaria. Total = 138** | | |
| Sources of information about the current (2015) guideline: | | |
| Had a copy of the guideline | 36 | 26.1 |
| Colleagues | 49 | 35.5 |
| Seminar/clinical meetings | 59 | 42.8 |
| Trainings/continuing medical education (CME) | 22 | 15.9 |
| Internet | 8 | 5.8 |

MRDT = Malaria rapid diagnostic test. ACTs = Artemisinin-based combination therapies.

having uncomplicated malaria should receive treatment based on confirmation of diagnosis with microscopy or MRDT; only 15 (9.6%) knew the results of parasitological diagnosis of uncomplicated malaria should be available within two hours of patients presenting; 84 (53.8%) knew anti-malarial drugs should be limited to only test positive cases; 111 (71.1%) knew treatment based solely on clinical suspicion should only be considered in children aged below five

years when microscopy or MRDT was not accessible; 154 (98.7%) Knew the recommended anti-malarial drug for uncomplicated malaria was ACTs; 149 (95.5%) knew the ACT of first choice was artemether-lumefantrine; 131 (84.0) knew ACTs were the recommended anti-malarial drugs for lactating mothers with uncomplicated malaria; 113 (72.4%) knew ACTs were the recommended anti-malarial drugs for uncomplicated malaria in second and third trimesters of pregnancy; 48 (30.8%) knew quinine plus clindamycin for 7 days was the first choice anti-malarial treatment for uncomplicated malaria in first trimester of pregnancy.

As shown (Table 2): 69 (44.2%) knew all patients suspected of having severe malaria should have a parasitological diagnosis before treatment; 100 (64.1%) knew parenteral artesunate was the recommended (first choice) anti-malarial drug for severe malaria; 73 (46.8%) knew that, once started, parenteral anti-malarial drugs for severe malaria should be given for at least 24 hours; 142 (91.0%) knew the patient should be given the full dose of an ACT after parenteral anti-malarial drug for severe malaria; 53 (34.0%) knew parenteral artesunate was the recommended anti-malarial drug for severe malaria in first trimester of pregnancy.

Overall, while 19 (12.2%) had good knowledge of the 2015 National Guidelines for Diagnosis and Treatment of Malaria, 110 (70.5%) had fair knowledge, and 27 (17.3%) had poor knowledge (Fig 1).

## Attitude toward the 2015 National Guidelines for Diagnosis and Treatment of Malaria

Among the 156 medical doctors (Table 3): 55 (35.3%) strongly agreed that all suspected cases of uncomplicated malaria should receive treatment based on parasitological diagnosis (with microscopy or MRDT) while 59 (37.8%) agreed, 7 (4.5%) were undecided, 23 (14.7%) disagreed, and 12 (7.7%) strongly disagreed. More than a quarter, 28 (17.9%) strongly agreed that patients with negative microscopy results should not be given anti-malarial drugs, 39 (25.0%) agreed, 19 (12.2%)were undecided, 58 (37.2%) disagreed, and 12 (7.7%) strongly disagreed. Only 9 (5.8%) strongly agreed that patients with negative MRDT results should not be given anti-malarial drugs while 23 (14.7%) agreed, 23 (14.7%) were undecided, 82 (52.6%) disagreed, and 19 (12.2%) strongly disagreed. Majority of them, 99 (63.5%) strongly agreed that the anti-malarial drug of choice was ACTs, 52 (33.3%) agreed, 1 (0.6) was undecided, 2 (1.3%) disagreed, and 2 (1.3%) strongly disagreed.

Majority of the doctors, 62 (39.7%) strongly agreed that parenteral anti-malarial drugs for severe malaria should be given for a minimum of 24 hours once commenced, 55 (35.3%) agreed, 9 (5.8%) were undecided, 24 (15.4%) disagreed, and 6 (3.8%) strongly disagreed. Majority of the doctors, 84 (53.8%), strongly agreed that after parenteral anti-malarial drugs for severe malaria the patient should be given the full dose of an ACT, 52 (33.3%) agreed, 12 (7.7%) were undecided, 4 (2.6%) disagreed, and 4 (2.6%) strongly disagreed. The medical doctors' attitude toward the recommendation that Sulphadoxine-pyremethamine and Chloroquine should not be used to treat malaria is as shown (Table 3). About a quarter of the medical doctors, 38 (24.4%) had good attitude toward the 2015 National Guidelines for Diagnosis and Treatment of Malaria, 74 (47.4%) had fair attitude, and 44 (28.2%) had poor attitude. About a fifth, 31 (19.9%) had good knowledge and attitude, 85 (54.5%) had fair knowledge and attitude, and 40 (25.6%) had poor knowledge and attitude (Fig 1).

## Practice of the 2015 National Guidelines for Diagnosis and Treatment of Malaria

Among the 156 medical doctors (Table 4): 8 (5.1%) never, 28 (17.9%) rarely, 76 (48.7%) sometimes, 38 (24.4%) often, and 6 (3.9%) always used only clinical features (presumptive

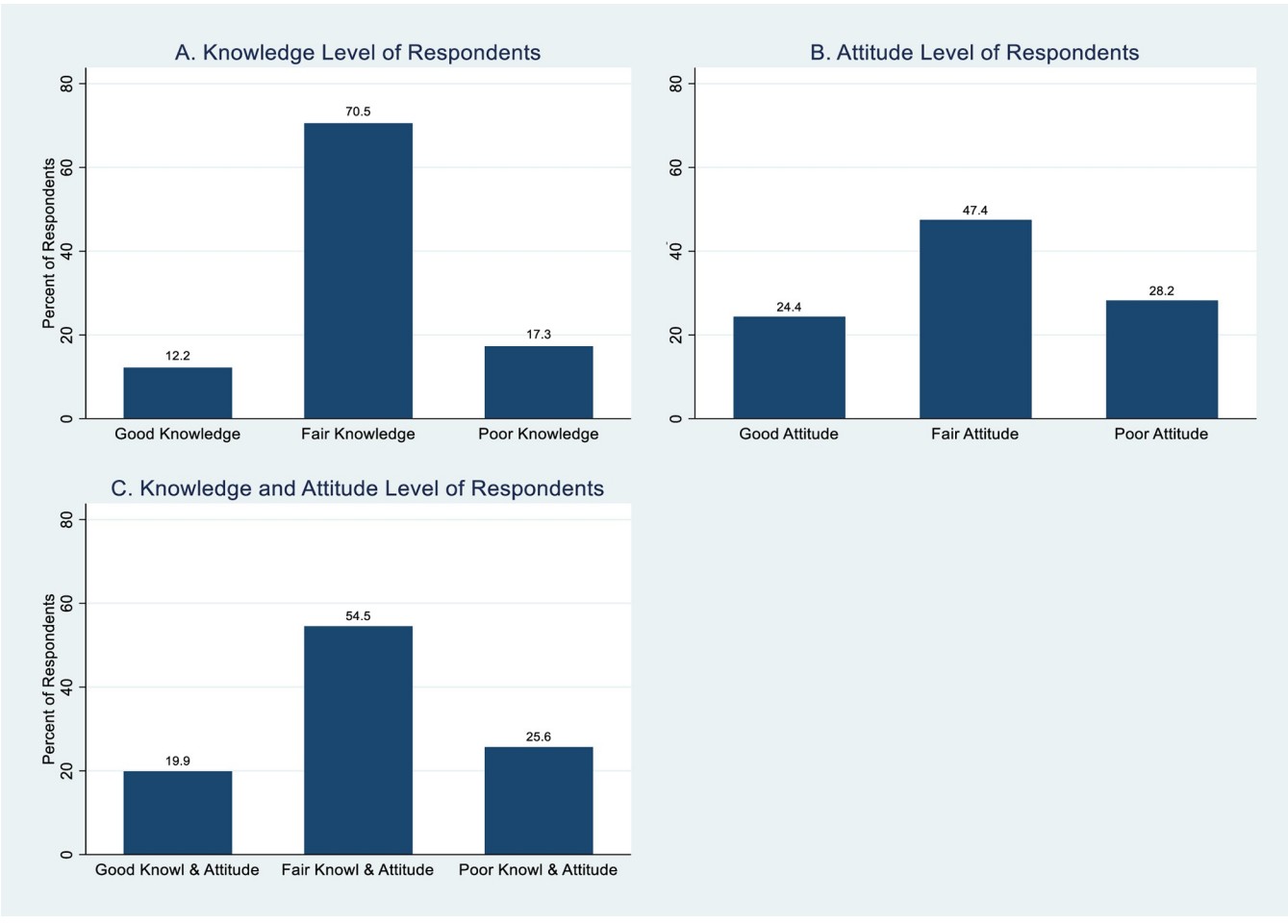

**Fig 1. Knowledge, attitude, and knowledge and attitude levels of respondents toward the 2015 National Guidelines for Diagnosis and Treatment of Malaria.**

diagnosis) as the basis for treating uncomplicated malaria; 24 (15.4%) never, 46 (29.5%) rarely, 70 (44.9%) sometimes, 12 (7.7%) often, and 4 (2.5%) always prescribed anti-malarial drugs for patients with negative microscopy test results; 13 (8.3%) never, 30 (19.2%) rarely, 92 (58.0%) sometimes, 16 (10.3%) often, and 5 (3.2%) always prescribed anti-malarial drugs for patients with negative MRDT results; and 115 (73.7%) always, 31 (19.9%) often, and 10 (6.4%) sometimes used ACTs to treat uncomplicated malaria. As shown (Table 4), 17 (10.9%) never, 40 (25.6%) rarely, 64 (41.0%) sometimes, 29 (18.6%) often, and 6 (3.9%) always used only clinical features (presumptive diagnosis) as the basis for treating severe malaria; 93 (59.6%) always, 47 (30.1%) often, and 16 (10.3%) sometimes prescribed parenteral anti-malarial drugs for severe malaria; and 93 (59.6%) always, 40 (25.6%) often, 13 (8.3%) sometimes, 6 (3.9%) rarely, and 4 (2.6%) never prescribed the full dose of an ACT after parenteral anti-malarial drugs for severe malaria. Among the doctors, 66 (42.3%) never, 52 (33.3%) rarely, 30 (19.2%) sometimes, 5 (3.2%) often, and 3 (1.9%) always prescribed sulphadoxine-pyremethamine for uncomplicated malaria while 88 (56.4%) never, 37 (23.7%) rarely, 25 (16.0%) sometimes, 3 (1.9%) often, and (1.9%) always prescribed chloroquine for uncomplicated malaria.

Overall, 59 (37.8%) had good practice of the 2015 National Guidelines for Diagnosis and Treatment of Malaria, 90 (57.7%) had fair practice, and 7 (4.5%) had poor practice (Fig 2).

**Table 3. Attitude of respondents toward the 2015 National Guidelines for Diagnosis and Treatment of Malaria.**

| Opinion items | Responses n (%), Total = 156 | | | | |
|---|---|---|---|---|---|
| | Strongly Agreed | Agreed | Un-decided | Disagreed | Strongly Disagreed |
| All suspected cases of uncomplicated malaria should receive treatment based on parasitological diagnosis (with microscopy or MRDT) | 55 (35.3) | 59 (37.8) | 7 (4.5) | 23 (14.7) | 12 (7.7) |
| Only patients with positive (microscopy or MRDT) test results should be given anti-malarial drugs | 36 (23.1) | 38 (24.3) | 10 (6.4) | 58 (37.2) | 14 (9.0) |
| Patients with negative microscopy results should not be given anti-malarial drugs | 28 (17.9) | 39 (25.0) | 19 (12.2) | 58 (37.2) | 12 (7.7) |
| Patients with negative MRDT results should not be given anti-malarial drugs | 9 (5.8) | 23 (14.7) | 23 (14.7) | 82 (52.6) | 19 (12.2) |
| Treatment based solely on clinical features should only be considered in children aged below five years when microscopy or MRDT is not accessible | 45 (28.9) | 57 (36.5) | 15 (9.6) | 32 (20.5) | 7 (4.5) |
| The anti-malarial drug of choice is ACTs | 99 (63.5) | 52 (33.3) | 1 (0.6) | 2 (1.3) | 2 (1.3) |
| The ACT of choice is artemether-lumefantrine while artesunate-amodiaquine is the alternate ACT | 78 (50.0) | 67 (42.9) | 2 (1.3) | 6 (3.9) | 3 (1.9) |
| All suspected cases of severe malaria should receive parasitological diagnosis (with microscopy or MRDT) before treatment | 37 (23.7) | 36 (23.1) | 8 (5.1) | 49 (31.4) | 26 (16.7) |
| Parenteral anti-malarial drugs for severe malaria should be given for a minimum of 24 hours once commenced | 62 (39.7) | 55 (35.3) | 9 (5.8) | 24 (15.4) | 6 (3.8) |
| After parenteral anti-malarial drugs for severe malaria the patient should be given the full dose of an ACT | 84 (53.8) | 52 (33.3) | 12 (7.7) | 4 (2.6) | 4 (2.6) |
| Sulphadoxine-pyremethamine should only be used for IPT in pregnancy (not to treat malaria) | 72 (46.2) | 56 (35.9) | 10 (6.4) | 15 (9.6) | 3 (1.9) |
| Chloroquine should not be used to treat malaria | 33 (21.1) | 29 (18.6) | 29 (18.6) | 50 (32.1) | 15 (9.6) |

MRDT = Malaria rapid diagnostic test. ACTs = Artemisinin-based combination therapies.

The overall level of practice by background characteristics of respondents is also depicted (Figs 3 and 4). The proportion of respondents who had good practice was (Fig 3): highest among those who were more than 40 years old, 19/47 (40.4%) and least among those aged 30

**Table 4. Respondent's practice of the 2015 National Guidelines for Diagnosis and Treatment of Malaria.**

| Practice items | Frequency of practice n (%), Total = 156 | | | | |
|---|---|---|---|---|---|
| | Never | Rarely | Some-times | Often | Always |
| Used only clinical features (presumptive diagnosis) as the basis for treating uncomplicated malaria | 8 (5.1) | 28 (17.9) | 76 (48.7) | 38 (24.4) | 6 (3.9) |
| Prescribed anti-malarial drugs for patients with positive microscopy test results | 0 | 0 | 0 | 41 (26.3) | 115 (73.7) |
| Prescribed anti-malarial drugs for patients with positive MRDT results | 0 | 0 | 2 (1.3) | 57 (36.5) | 97 (62.2) |
| Prescribed anti-malarial drugs for patients with negative microscopy test results | 24 (15.4) | 46 (29.5) | 70 (44.9) | 12 (7.7) | 4 (2.5) |
| Prescribed anti-malarial drugs for patients with negative MRDT results | 13 (8.3) | 30 (19.2) | 92 (58.0) | 16 (10.3) | 5 (3.2) |
| Used ACTs to treat uncomplicated malaria | 0 | 0 | 10 (6.4) | 31 (19.9) | 115 (73.7) |
| Used artemether-lumefantrine to treat uncomplicated malaria | 2 (1.3) | 1 (0.6) | 11 (7.0) | 53 (33.9) | 89 (57.0) |
| Used artesunate-amodiaquine to treat uncomplicated malaria | 4 (2.6) | 33 (21.2) | 45 (28.8) | 47 (30.1) | 27 (17.3) |
| Used only clinical features (presumptive diagnosis) as the basis for treating severe malaria | 17 (10.9) | 40 (25.6) | 64 (41.0) | 29 (18.6) | 6 (3.9) |
| Prescribed parenteral anti-malarial drugs for severe malaria | 0 | 0 | 16 (10.3) | 47 (30.1) | 93 (59.6) |
| Prescribed the full dose of an ACT after parenteral anti-malarial drugs for severe malaria | 4 (2.6) | 6 (3.9) | 13 (8.3) | 40 (25.6) | 93 (59.6) |
| Prescribed sulphadoxine-pyremethamine for uncomplicated malaria | 66 (42.3) | 52 (33.3) | 30 (19.2) | 5 (3.2) | 3 (1.9) |
| Prescribed chloroquine for uncomplicated malaria | 88 (56.4) | 37 (23.7) | 25 (16.0) | 3 (1.9) | 3 (1.9) |

MRDT = Malaria rapid diagnostic test. ACTs = Artemisinin-based combination therapies.

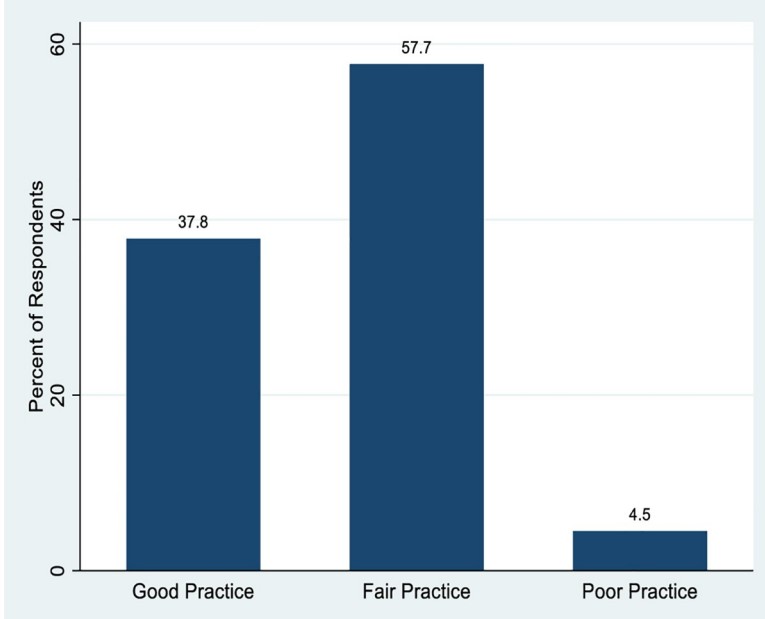

**Fig 2. Respondents' overall level of practice of the 2015 National Guidelines for Diagnosis and Treatment of Malaria.**

years and below, 9/26 (34.6%); higher among females, 20/48 (41.7%) compared with males, 39/108(36.1%); highest among the senior registrars, 33/67 (49.3%), and least among the medical officers, 6/27 (22.2%); and highest among those who had 11–15 years of practicing experience, 13/30 (43.3%), and least among those who had 5or less years of practicing experience, 12/37 (32.4%).

The proportion of respondents who had good practice was (Fig 4): highest among those who had good knowledge, 11/19 (57.9%), and least among those who had poor knowledge, 6/27 (22.2%); highest among those who had good attitude, 26/38 (68.4%), and least among those who had poor attitude, 9/44 (20.5%); higher among those whose primary place of practice was tertiary health facility (federal university teaching hospital), 54/134 (40.3%) compared with those in secondary health facilities, 5/22 (22.7%). Among those in the federal university teaching hospital, the doctors in community medicine department had the highest proportion with good practice, 18/34 (52.9%) while those in the obstetrics and gynaecology department had the least proportion with good practice, 9/28 (32.1%).

## Factors associated with good practice of the 2015 National Guidelines for Diagnosis and Treatment of Malaria

The associations between background factors and good practice of the 2015 National Guidelines for Diagnosis and Treatment of Malaria among the medical doctors are presented in Table 5. For each factor, crude and adjusted odds ratios and their respective 95% CI and p-values are reported for each of the other categories compared to a reference category. The crude and adjusted p-values of the overall effect of each factor are also reported. Only the attitude level of the medical doctors toward the 2015 National Guidelines for Diagnosis and Treatment of Malaria was a predictor of (significantly associated with) good practice (adjusted p-value of overall effect = 0.0003) (Table 5).

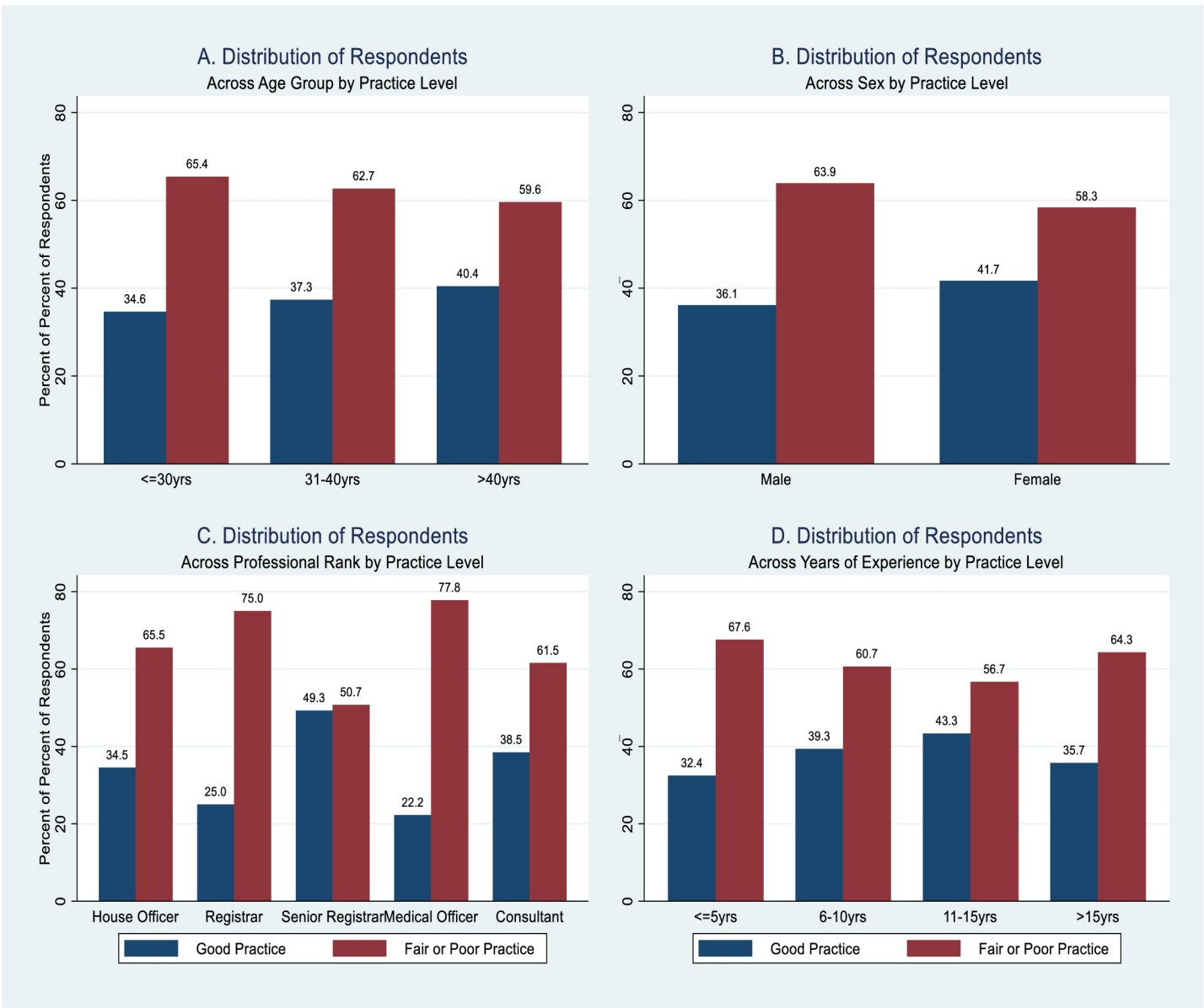

**Fig 3. Respondents' overall level of practice of the 2015 National Guidelines for Diagnosis and Treatment of Malaria by background characteristics.**

## Discussion

This study evaluated the knowledge, attitude, and practice of the 2015 National Guidelines for Diagnosis and Treatment of Malaria, as well as associated factors, among medical doctors in Ebonyi state, Nigeria. It was found that overall, only 12.2% of the medical doctors had good knowledge of the Guidelines, while 70.5% had fair knowledge and 17.3% had poor knowledge. Specifically, 88.5% were aware there was a national guideline for diagnosis and treatment of malaria. A similar but slightly higher value was reported by a study in Ogun state, Nigeria [14] where 96.6% of health workers (98.2% of public health workers and 94.9% of private health workers) were aware of the national malaria treatment guidelines. Similar finding was reported by a study in Tanzania [16] where 96.4% of healthcare workers were aware of malaria

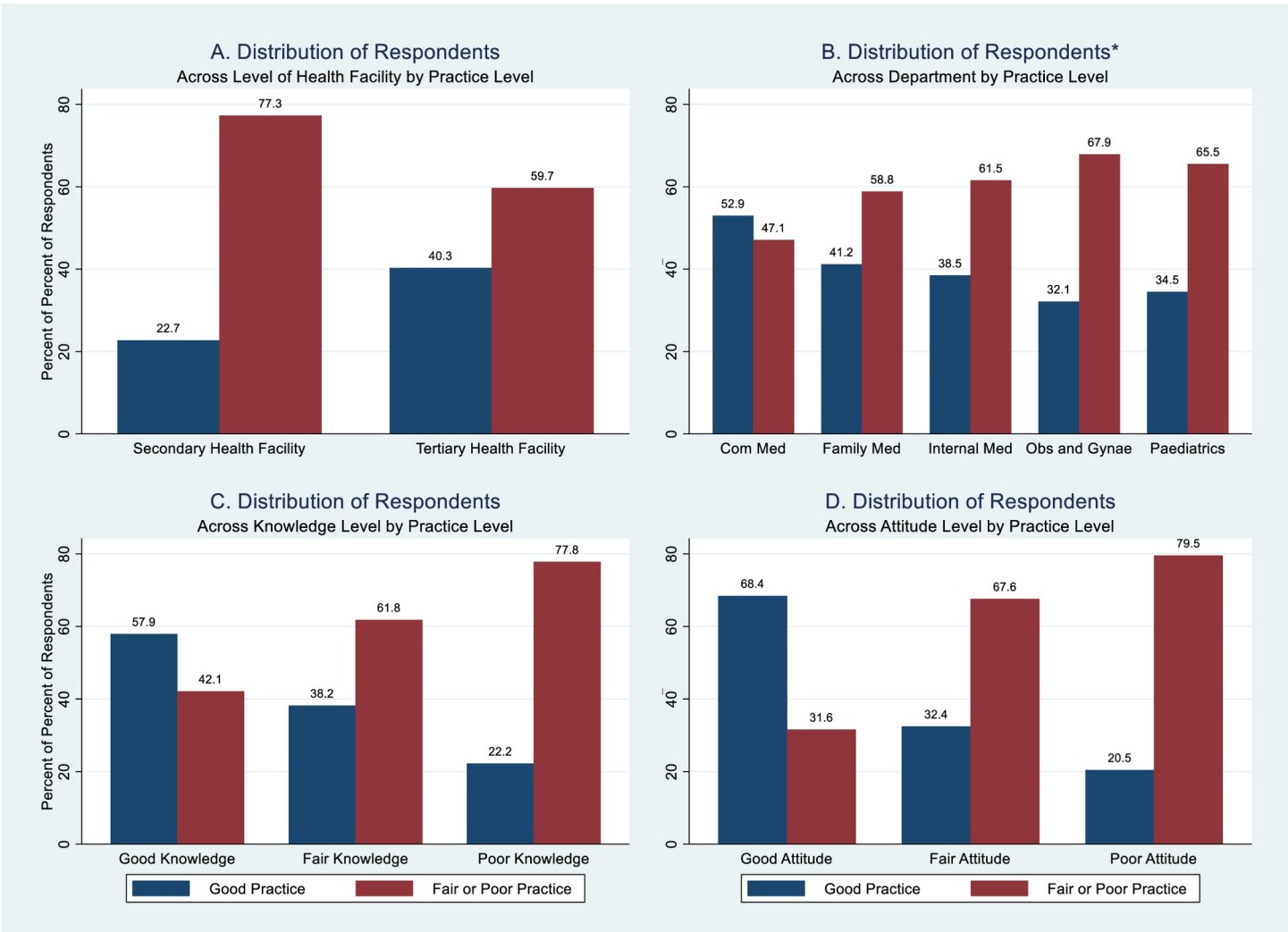

**Fig 4. Respondents' overall level of practice of the 2015 National Guidelines for Diagnosis and Treatment of Malaria by background characteristics.** *Respondents in Tertiary Health Facility (federal university teaching hospital).

diagnosis and treatment guidelines. Also, in a study in Western Kenya [15], 98.7% of healthcare providers in health facilities and 27.0% of those in drug outlets (75% overall) were aware of the national malaria treatment guidelines. In another study in Equatorial Guinea [13], 42.3% of the healthcare workers at health centres and 66.7% of those at hospitals (52.3% overall) were aware of the national malaria treatment guidelines.

The above findings indicate that the awareness of the existence of national malaria treatment guidelines is high among healthcare workers in Africa. However, in this study, despite the high level of awareness among the medical doctors, the knowledge of the salient recommendations of the 2015 Guidelines is rather low. This low level of knowledge might have resulted from the fact that only about half of the medical doctors surveyed had seen the 2015 Guidelines while less than half had read the Guidelines. Also, only about a fourth of those who were aware there was a national guideline (less than a fourth of all those surveyed) had a copy of the 2015 Guidelines. Although medical doctors have several sources for learning about optimal management of malaria patients, including clinical meetings or seminars, it is expected

**Table 5. Factors associated with good practice of the 2015 National Guidelines for Diagnosis and Treatment of Malaria.**

| Factors | Good practice* | | | |
|---|---|---|---|---|
| | cOR (95% CI) | p-value | aOR (95% CI) | p-value |
| **Age group** (in years) | | 0.8795[A] | | 0.5253[A] |
| ≤30 | 1.0 | – | 1.0 | – |
| 31–40 | 1.1 (0.4–2.8) | 0.801 | 0.5 (0.1–3.6) | 0.524 |
| >40 | 1.3 (0.5–3.5) | 0.625 | 0.9 (0.1–9.5) | 0.980 |
| **Sex** | | | | |
| Male | 1.0 | – | 1.0 | – |
| Female | 1.3 (0.6–2.5) | 0.509 | 1.5 (0.6–3.5) | 0.347 |
| **Professional rank** | | 0.1001[A] | | 0.3451[A] |
| House officer | 1.0 | – | 1.0 | – |
| Registrar | 0.6 (0.2–2.3) | 0.481 | 0.5 (0.1–4.6) | 0.533 |
| Senior registrar | 1.8 (0.7–4.6) | 0.184 | 1.8 (0.2–19.9) | 0.625 |
| Medical officer | 0.5 (0.2–1.8) | 0.313 | 0.6 (0.02–18.1) | 0.786 |
| Consultant | 1.2 (0.3–4.6) | 0.804 | 0.7 (0.04–9.7) | 0.758 |
| **Duration of practice** (in years) | | 0.8126[A] | | 0.9040[A] |
| ≤5 | 1.0 | – | 1.0 | – |
| 6–10 | 1.4 (0.6–3.2) | 0.492 | 0.9 (0.1–8.1) | 0.970 |
| 11–15 | 1.6 (0.6–4.3) | 0.360 | 0.7 (0.1–7.8) | 0.781 |
| >15 | 1.2 (0.4–3.3) | 0.782 | 1.2 (0.1–13.3) | 0.901 |
| **Level of current primary place of practice** | | | | |
| Secondary health facility[‖] | 1.0 | – | 1.0 | – |
| Tertiary health facility[‖‖‖] | 2.3 (0.8–6.6) | 0.123 | 0.8 (0.1–11.5) | 0.874 |
| **Knowledge level**[**] | | 0.0567[A] | | 0.3760[A] |
| Poor | 1.0 | – | 1.0 | – |
| Fair | 2.2 (0.8–5.8) | 0.125 | 1.7 (0.6–5.1) | 0.341 |
| Good | 4.8 (1.3–17.4) | 0.017 | 2.8 (0.7–12.3) | 0.162 |
| **Attitude level**[***] | | 0.0001[A] | | 0.0003[A] |
| Poor | 1.0 | – | 1.0 | – |
| Fair | 1.9 (0.8–4.5) | 0.164 | 1.7 (0.7–4.5) | 0.248 |
| Good | 8.4 (3.1–22.9) | <0.001 | 8.6 (2.8–25.8) | <0.001 |

cOR = Crude odds ratio from bivariate logistic regression. aOR = Adjusted odds ratio from multivariate logistic regression.

[A]p-value from overall test for the effects of the independent variables (using the postestimation test command in Stata).

[‖]Comprised of private hospitals, missionary hospitals, general hospitals, and federal polytechnique medical centre.

[‖‖‖]Federal university teaching hospital.

*Level of practice of the 2015 national guidelines for diagnosis and treatment of malaria (practice score >75% of the total of 65 is good, < = 75% is fair or poor).

**Level of knowledge about the 2015 national guidelines for diagnosis and treatment of malaria (knowledge score >75% of the total of 23 is good, 50–75% is fair, <50% is poor).

***Level of attitude towards the 2015 national guidelines for diagnosis and treatment of malaria (attitude score >75% of the total of 60 is good, 50–75% is fair, <50% is poor).

that having a copy of the National Guidelines (and regularly using it as reference material) would enhance comprehensive and in-depth knowledge of such Guidelines.

In this study, with regards to the overall attitude level, while 24.4% of the medical doctors had good attitude toward the 2015 National Guidelines for Diagnosis and Treatment of Malaria, 47.4% had fair attitude, and 28.2% had poor attitude. Specifically 35.3% strongly agreed and 37.8% agreed (73.1% agreed or strongly agreed) that all suspected cases of uncomplicated malaria should receive treatment based on parasitological diagnosis (with microscopy

or MRDT). A slightly lower value was reported by a study among medical doctors in Enugu state university teaching hospital [11], where 64.0% of the medical doctors thought it was important to confirm a diagnosis with a parasitological test before commencing treatment for malaria.

In this study, with regards to the overall practice level, 37.8% had good practice of the 2015 National Guidelines for Diagnosis and Treatment of Malaria, 57.7% had fair practice, and 4.5% had poor practice. Specifically, 5.1% never, 17.9% rarely, 48.7% sometimes, 24.4% often, and 3.9% always used only clinical symptoms as the basis for treating uncomplicated malaria. Largely similar findings were reported by a study in Ogun state, Nigeria [14] where 44.1% of health workers strictly adhered to the national malarial treatment guidelines by use of the recommended anti-malarial drugs following parasitological confirmation of cases, 32.7% partially adhered by use of the recommended anti-malarial drugs without parasitological confirmation, and 22.5% did not adhere by non-useof the recommended anti-malarial drugs in the absence of parasitological confirmation. Also, a study in Tanzania [16] reported that 54.6% of healthcare workers adhered strictly, 5.1% adhered partially, and 40.3% did not adhere to the diagnosis and treatment guidelines.

In another study among medical doctors in Enugu state university teaching hospital [11], 40.7% of the medical doctors usually made diagnosis of malaria based on only clinical symptoms. A study in Uganda [20] reported 50.6% prevalence of adherence to national malaria treatment guidelines among public healthcare workers as 50.6% of patients suspected of having malaria received parasitological diagnosis and malaria treatment (with recommended anti-malarial drugs) for only positive results. Another study in Equatorial Guinea [17] reported that 80.2% of the diagnoses of uncomplicated malaria (in children less than 15 years of age) made by healthcare workers were based on parasitological confirmation.

Although the above studies used different definitions of adherence by healthcare workers and different outcome measures, the findings broadly indicate sub-optimal practice or adherence to diagnostic and treatment guidelines in Africa. In this study, even though most of the medical doctors were aware of existence of a national guidelines for diagnosis and treatment of malaria which was readily available and accessible, and despite the fact that most were in an academic environment (teaching hospital), only just over a third of them had good practice of the 2015 Guidelines. The reason for this low level of practice might be related to the fact that, due to their higher level of training and clinical judgement, many medical doctors often tend to believe in and to rely on their clinical discretion when taking certain decisions regarding patient management without recourse to written guidelines. Not surprisingly, the attitude level of the medical doctors toward the 2015 National Guidelines for Diagnosis and Treatment of Malaria was found to be a predictor of good practice.

The fact that this study collected self-reported data about practices was a limitation due to the possibility of reporting bias as there might be the tendency for respondents to overestimate desirable outcomes and underestimate undesirable outcomes. However, the bias was minimized by making the questionnaire anonymous and by assuring respondents of a high degree of confidentiality.

## Conclusions

Although most of the medical doctors in Ebonyi state, Nigeria, were aware of the existence of a national guideline for diagnosis and treatment of malaria, only about one in eight had good knowledge, about a quarter had good attitude, and just over a third had good practice of the 2015 National Guidelines for Diagnosis and Treatment of Malaria. The attitude level of the medical doctors toward the 2015 Guidelines was a predictor of good practice. The findings

highlight the necessity of policy interventions such as training and re-training of the medical doctors in Ebonyi state, Nigeria, to improve their attitude toward the National Guidelines for Diagnosis and Treatment of Malaria in order to optimize the management of suspected malaria patient and malaria control.

## Supporting information

**S1 Questionnaire.**
(DOCX)

**S1 Dataset.**
(XLS)

## Author Contributions

**Conceptualization:** Ugwu I. Omale.

**Data curation:** Ugwu I. Omale.

**Formal analysis:** Ugwu I. Omale.

**Investigation:** Ugwu I. Omale.

**Methodology:** Ugwu I. Omale.

**Project administration:** Ugwu I. Omale.

**Resources:** Ugwu I. Omale.

**Validation:** Ugwu I. Omale.

**Writing – original draft:** Ugwu I. Omale.

**Writing – review & editing:** Ugwu I. Omale.

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
