## [Decision Letter · Decision Letter 0]

19 Apr 2021

PONE-D-21-07204

Knowledge, attitude, and practice of the National Guidelines for Diagnosis and Treatment of Malaria among medical doctors in Ebonyi state, Nigeria: a cross-sectional survey

PLOS ONE

Dear Dr. Omale,

Thank you for submitting your manuscript to PLoS ONE. After careful consideration, we felt that your manuscript requires revision, following which it can possibly be reconsidered. Although your manuscript was of interest to the three  reviewers, major concerns were related to study design, data analysis and results.  According to the reviewers, the methods were not described in enough details to allow suitably skilled investigators to fully replicate and evaluate the study. In addition, a significant number of issues should be clarified and/or adjust, particularly, in the methods, otherwise the MS’s results may be compromised. Finally, MS should be submitted to a professional proofreading process.

Please submit your revised manuscript by May 16.  If you will need more time than this to complete your revisions, please reply to this message or contact the journal office at plosone@plos.org. Please include the following items when submitting your revised manuscript:

We look forward to receiving your revised manuscript.

Kind regards,

Luzia Helena Carvalho, Ph.D.

Academic Editor

PLOS ONE

Journal Requirements:

Please include additional information regarding the survey or questionnaire used in the study and ensure that you have provided sufficient details that others could replicate the analyses. For instance, if you developed a questionnaire as part of this study and it is not under a copyright more restrictive than CC-BY, please include a copy, in both the original language and English, as Supporting Information. Moreover, please include more details on how the questionnaire was pre-tested, and whether it was validated.

Please provide additional details regarding participant consent. In the ethics statement in the Methods and online submission information, please ensure that you have specified how verbal consent was documented and witnessed.

4a) If there are ethical or legal restrictions on sharing a de-identified data set, please explain them in detail (e.g., data contain potentially identifying or sensitive patient information) and who has imposed them (e.g., an ethics committee). Please also provide contact information for a data access committee, ethics committee, or other institutional body to which data requests may be sent.

4b) If there are no restrictions, please upload the minimal anonymized data set necessary to replicate your study findings as either Supporting Information files or to a stable, public repository and provide us with the relevant URLs, DOIs, or accession numbers. Please see http://www.bmj.com/content/340/bmj.c181.long for guidelines on how to de-identify and prepare clinical data for publication. For a list of acceptable repositories, please see http://journals.plos.org/plosone/s/data-availability#loc-recommended-repositories.

Reviewers' comments:

Reviewer's Responses to Questions

**Comments to the Author**

1. Is the manuscript technically sound, and do the data support the conclusions?

Reviewer #1: Yes

Reviewer #2: Yes

Reviewer #3: Yes

2. Has the statistical analysis been performed appropriately and rigorously? 

Reviewer #1: Yes

Reviewer #2: Yes

Reviewer #3: Yes

3. Have the authors made all data underlying the findings in their manuscript fully available?

Reviewer #1: Yes

Reviewer #2: Yes

Reviewer #3: No

4. Is the manuscript presented in an intelligible fashion and written in standard English?

Reviewer #1: Yes

Reviewer #2: Yes

Reviewer #3: Yes

5. Review Comments to the Author

Reviewer #1: Comments to the author.

The article is very important study about malaria management in Ebonyi state, Nigeria and reflect what happen in African countries.

The paper is very clear and data very explored. Minor reviews on the writing is desired.

Abstract

Line 32 - knowledge, attitude, and practice respectively for knowledge, attitude, and practice scores of <50% - delete the repeated words: knowledge, attitude, and practice and review the sentence;

Line 39 - Overall, 59 (37.8%) had good… Review. Change “Overall”.

Line 42 - predictor of good practice of the 2015 Guidelines (adjusted p-value of overall effect=0.0003). Eliminate the “of the 2015 Guidelines”.

Introduction

Line 57: There were several recommendations in the 2015 guidelines. Eliminate the 2015.

Line 69: have the main responsibility for treating patients with both uncomplicated and severe malaria. Eliminate “both uncomplicated and severe”.

Line 71: knowledge of the guideline, have good attitude towards the guideline, and above all, have to practice. Eliminate the repeated verb, “have”.

Methods

Line 91: This study was an analytical cross-sectional study. The study was carried out in May, 2019 in Ebonyi state. Change to: “This study was an analytical cross-sectional survey carried out in May, 2019 in Ebonyi state” or other similar, to simplify the phrase.

Line 120 : Data was entered using Microsoft Excel 2007 (Microsoft Inc., Redmond, Washington), and was verified and… substitute the and was by a common .

Results

Line 152: Majority of 108 (69.2%) were males and 48 (30.8%) were… Correct to Majority of them, 108 (69.2%) were males….

Line 155: and were in Community Medicine Department in the teaching hospital, 34 (21.8%) (Table 1). Change the order to: …and 34 (21.8%) were in Community…

Page 9 - Line 185 to Line 201: Some phrases started by While… Please review it. May changing the word by other to give more sense to your results.

Line 222 : While 66 (42.3%) never, 52 (33.3%) rarely, 30 (19.2%) sometimes, 5 (3.2%) often, and 3 (1.9%) always prescribed… Change While… Start the phase to another word.

Line 226: Overall, while 59 (37.8%) had good practice of the 2015 National Guidelines for Diagnosis and Treatment of… Cut while.

Line 329: The fact that this study collected self-reported data about practices was a limitation because of due the possibility of… Change the word because of… It gives more sense.

Reviewer #2: Review comments

General comment

This is an interesting manuscript and well written. I commend the authors for a good work. Even though the sample size for the study is small, the relevance of the study cannot be in doubt.

Before this manuscript could be accepted for publication, Authors should explain the selection of the respondents. (Responses to the questions raised in this regard will be necessary). The authors should also take another look on the scoring of the attitude and practice variables or provide a reference to what they have done.

An explanation is also required on why all the variables merited inclusion in the multivariate logistic regression model.

Abstract

Methods

How were the medical doctors selected for inclusion in the study?

How many variables were used to assess the outcome variables?

Why categorize the outcome variables into three categories when authors had in mind to perform logistic regression analysis?

Authors should state the level of statistical significance.

Results

Authors should explain the meaning of adjusted p value

Present logistic regression analysis results using adjusted odds ratio and 95% confidence interval

The outcome variable should be well described. Is it good knowledge of what?

Conclusions

What are the public health implications of the study findings?

How could the attitude of the medical doctors towards the National Guideline be improved?

Introduction

My commendation to the authors for a well written introduction.

Line 82 appear like a stand-alone statement. To my knowledge…… This should be reviewed.

Methods

Study design and setting

The description of the study as ‘analytical cross-sectional study’ should be reviewed.

Study participants and data collection

The selection of the respondents is not very clear

Were all the medical doctors present on the day of data collection in the departments/units as stated in the manuscript included in the study?

Then if it was a total population study, what is the response rate.

The authors should give the readers an idea of the total number of doctors working in those areas where malaria cases are managed? As a follow up to that a good description of how the respondents were selected should be done.

Who designed the questionnaire and where was it pre-tested?

Statistical analysis

It will be good to state the number of variables used to assess the attitude of the respondents to the National Guideline. The scoring of the variables used to assess attitude of the respondents is not very clear. What is the rationale for awarding two marks to someone who is undecided on a particular variable? In any case, the authors should provide a reference that justifies this very decision.

Do the same for the scoring of the variables used to assess practice level of the respondents.

How was the practice level dichotomized?

Results

A proportion of 43% cannot be regarded as majority. Use highest proportion instead. Correct in other areas. Majority is recorded for proportions above 50%.

Table 1

Second column should be frequency (n=156)

Remove the row total (Do the same for other tables).

Figures 3A and 3B should be removed. Then for table 5, remove the crude odds ratios and replace with proportions that had good and poor practice. The p value for the bivariate analysis should be included. For the multivariate analysis, present the adjusted odds ratio and 95% confidence interval. The p value on multivariate analysis may be included.

Remove the constants in table 5

All the variables had both crude and adjusted odds ratios. What qualified the variables for inclusion in the multivariate logistic regression model? If all the variables merited inclusion as seen in the result, what is the rationale for that?

Considering the small sample size as seen in this study, why should duration of practice in years have 5 categories? This remarkably affected the adjusted odds ratio and should be reviewed. Review the number of categories for other independent variables.

Discussion

Include Nigeria after Ogun state for better understanding

Discuss the predictors of good practice of the National guideline

Conclusions

This should be in a single paragraph

Use predictor of good practice instead of strong predictor

Reviewer #3: Review Comments on a Manuscript Titled: Knowledge, attitude, and practice of the National Guidelines for Diagnosis and Treatment of Malaria among medical doctors in Ebonyi state, Nigeria: a cross-sectional survey

General comments

This is an interesting article that addresses an issue of importance in the control of Malaria. I However, the paper needs some work, particularly on data analyses, results section and Discussion. The article also has several grammatical errors particularly on the results section and so needs close proof reading to correct these or copy edit. Several specific issues are as highlighted below:

Title

- The title does not match with the study objective. The title reflects a study which is purely descriptive, however, the objective of the study is to also assess factors associated with practices regarding the Malaria guideline. Kindly recast the title to reflect that the study assessed and reports about predictors.

Introduction

- Line 66: ….the risk of drug resistance. It should be “…………risk for development of drug resistance”

- Line 87-88: The aim of this study was to evaluate the knowledge, attitude, and practice of the 2015 National Guidelines for Diagnosis and Treatment of Malaria, and associated factors, among medical doctors in Ebonyi state, Nigeria. Rephrase the title to match with the study aim.

Methodology

Study design and setting

The study design should be cross-sectional analytical design.

Study participants and data collection

The author states that they selected 156 medical doctors, what was the basis for selecting this number of participants, how sure were they that by engaging 156 medical doctors the study will be able to provide reliable answer to the questions it tries to answer?

Results

Sociodemographic and background characteristics

- When describing results, remove the word “of” after “the majority”, if there is a need of using the word “of” then it should be followed by “respondents” then number and percentage.

- Table 1 should be inserted after paragraph 1 that describes sociodemographic characteristics of respondents

- The way the total is indicated in table 1 (156) it as if 156 is the total of all the values for all the variables in the table. The best way is to indicate this total in the table title in brackets. This should be done for all the tables

- In table 1, on Place of practice as a variable should have two categories i.e. Secondary health facility and Federal university teaching hospital. This will make this table easier to comprehend. Then plot a bar chart for each of these categories to indicate the distribution of specific place of practices.

- To indicate less or equal use ≤ instead of <=

- The sum of the percentages of age categories gets to 100.1, please check

Knowledge of the 2015 National Guidelines for Diagnosis and Treatment of Malaria

- Table 2 should be cited at the end of the paragraph

- Totals should be on every major category and should be indicated in brackets, i.e. “Knowledge items (156)”, “Among those that were aware there was a national guideline for diagnosis and treatment of malaria (138)”.

- “Among those that were aware there was a national guideline for diagnosis and treatment of malaria” this can be removed and remain with “Sources of information on the national malaria treatment guideline”

- Because items in table 2 are multiple responses items, putting a 100 on the percentage as total is potentially misleading because when you sum up individual percentages goes far beyond 100.

Factors associated with good practice of the 2015 National Guidelines for Diagnosis and Treatment of Malaria

- Why have you included all variable in your multivariable model? Usually, one would construct this using the variables with a predefined p-value from the univariable analysis. What was the criteria for inclusion of variables into the multivariable model?

Discussion

- Authors repeats results description in the discussion section. Kindly revise the discussion section by discussing findings rather than rewriting the findings as it appears now.

Conclusions

- The authors recommend interventions that would improve attitude, it will be more meaningful to probably provide examples of such interventions.

6. PLOS authors have the option to publish the peer review history of their article (what does this mean?). If published, this will include your full peer review and any attached files.

Reviewer #1: **Yes: **Adilson DePINA

Reviewer #2: **Yes: **EDMUND NDUDI OSSAI

Reviewer #3: No

---

## [Author Response · Author response to Decision Letter 0]

15 Jul 2021

RESPONSE TO REVIEWERS

Reviewer # 1 

Abstract

Comment: Line 32 - knowledge, attitude, and practice respectively for knowledge, attitude, and practice scores of <50% - delete the repeated words: knowledge, attitude, and practice and review the sentence;

Response: Please note that each respondent was categorized as having poor, fair, or good knowledge, attitude, and practice respectively based on the knowledge, attitude, and practice scores of <50%, 50–75%, and >75%. “for” has been replaced with “based on” for clarity and this might be more understandable than when the second “knowledge, attitude, and practice” is deleted viz: Each respondent was categorized as having poor, fair, or good knowledge, attitude, and practice respectively for scores of <50%, 50–75%, and >75%. In this case, the query that might arise is, “which scores?” So I think it should be indicated straightway that is was the knowledge, attitude, and practice scores.

Comment: Line 39 - Overall, 59 (37.8%) had good… Review. Change “Overall”.

Response: “Overall” have been replaced with “Among the medical doctors” in line 37 and removed from line 39.

Comment: Line 42 - predictor of good practice of the 2015 Guidelines (adjusted p-value of overall effect=0.0003). Eliminate the “of the 2015 Guidelines”.

Response: The phrase has been deleted.

Introduction

Comments: 

Line 57: There were several recommendations in the 2015 guidelines. Eliminate the 2015.

Line 69: have the main responsibility for treating patients with both uncomplicated and severe malaria. Eliminate “both uncomplicated and severe”.

Line 71: knowledge of the guideline, have good attitude towards the guideline, and above all, have to practice. Eliminate the repeated verb, “have”.

Response: The corrections have been effected.

Methods

Comments: 

Line 91: This study was an analytical cross-sectional study. The study was carried out in May, 2019 in Ebonyi state. Change to: “This study was an analytical cross-sectional survey carried out in May, 2019 in Ebonyi state” or other similar, to simplify the phrase.

Line 120 : Data was entered using Microsoft Excel 2007 (Microsoft Inc., Redmond, Washington), and was verified and… substitute the and was by a common .

Response: The corrections have been effected.

Results

Comment: Line 152: Majority of 108 (69.2%) were males and 48 (30.8%) were… Correct to Majority of them, 108 (69.2%) were males….

Response: The correction has been effected.

Comment: Line 155: and were in Community Medicine Department in the teaching hospital, 34 (21.8%) (Table 1). Change the order to: …and 34 (21.8%) were in Community…

Response: The starting phrase “The majority of the respondents:” applies to every other phrase in the compound sentence. Changing the order as suggested will alter the flow of the compound sentence.

Comments: 

Page 9 - Line 185 to Line 201: Some phrases started by While… Please review it. May changing the word by other to give more sense to your results.

Line 222 : While 66 (42.3%) never, 52 (33.3%) rarely, 30 (19.2%) sometimes, 5 (3.2%) often, and 3 (1.9%) always prescribed… Change While… Start the phase to another word.

Line 226: Overall, while 59 (37.8%) had good practice of the 2015 National Guidelines for Diagnosis and Treatment of… Cut while.

Line 329: The fact that this study collected self-reported data about practices was a limitation because of due the possibility of… Change the word because of… It gives more sense.

Response: The corrections have been effected.

Reviewer # 2 

Abstract

Methods

Comments: 

How were the medical doctors selected for inclusion in the study?

How many variables were used to assess the outcome variables?

Response: They were selected by convenience sampling as stated in the body of the manuscript under the ”study participants and data collection” section. The numbers of variables used for the assessment have been added to the body of the manuscript. There is no space to include other details in the Abstract section due to the Journal’s word count.

Comment: Why categorize the outcome variables into three categories when authors had in mind to perform logistic regression analysis?

Response: Categorization the outcome variables into three categories are to convey more detailed information for the descriptive aspect of the analysis. This does not prevent dichotomization of the practice variable for inferential analysis.

Comment: Authors should state the level of statistical significance. 

Response: The suggestion has been effected.

Results

Comment: Authors should explain the meaning of adjusted p value 

Response: Just like adjusted odds ratio, adjusted p-value is p-value from adjusted analysis (multivariate analysis). It is not necessary to be explaining this in the manuscript. 

Comment: Present logistic regression analysis results using adjusted odds ratio and 95% confidence interval 

Response: For multi-level variable such as attitude level with three categories, the overall effect is preferably reported in the abstract. There is no space to present the pair-wise result as suggested.

Comment: The outcome variable should be well described. Is it good knowledge of what? Response: The outcome variables are knowledge, attitude, and practice. 

Conclusions

Comment: What are the public health implications of the study findings? 

Response: The implication of the main finding is as stated. The investigator cannot speculate on other implications.

Comment: How could the attitude of the medical doctors towards the National Guideline be improved? 

Response: It could be through training and re-training of the medical doctors.

Introduction

Comment: Line 82 appear like a stand-alone statement. To my knowledge…… This should be reviewed. 

Response: The investigator is the only author of the manuscript and seems not to understand this comment.

Methods

Study design and setting

Comment: The description of the study as ‘analytical cross-sectional study’ should be reviewed.

Response: The study was actually an analytical cross-sectional study

Study participants and data collection

Comments: 

The selection of the respondents is not very clear

Were all the medical doctors present on the day of data collection in the departments/units as stated in the manuscript included in the study?

Then if it was a total population study, what is the response rate.

The authors should give the readers an idea of the total number of doctors working in those areas where malaria cases are managed? As a follow up to that a good description of how the respondents were selected should be done.

Response: Convenience sampling was employed as stated in the manuscript.

Comment: Who designed the questionnaire and where was it pre-tested? 

Response: The questionnaire was designed by the investigator and pre-tested among doctors who were later not included in the study.

Statistical analysis

Comments: 

It will be good to state the number of variables used to assess the attitude of the respondents to the National Guideline. The scoring of the variables used to assess attitude of the respondents is not very clear. What is the rationale for awarding two marks to someone who is undecided on a particular variable? In any case, the authors should provide a reference that justifies this very decision.

Do the same for the scoring of the variables used to assess practice level of the respondents.

Response: The numbers of variables have been added to the “study participants and data collection” section. The investigator believes of the scoring of attitude and practice is clearly described. Please note that, although many scholars might chose to respectively assign scores of 1–5 to the five responses, other scholars might chose to use different “weighted” scoring approach based on the concept of their study. For example, a study assigned scored of 1–4 to “strongly agree or disagree” or “somewhat agree or disagree” and “0” to “don’t know/can’t say” (Vaidya A, Aryal UR, Krettek A. Cardiovascular health knowledge, attitude and practice/behaviour in an urbanising community of Nepal: a population-based cross-sectional study from Jhaukhel-Duwakot Health Demographic Surveillance Site. BMJ Open 2013;3:e002976).

In this study, a participant who was “undecided” regarding a salient recommendation in a diagnosis and treatment guideline should receive a higher score than another who “disagreed” or “strongly disagreed”. However, the interval between scores is not a “mathematical derivation”.

Assigning the middle response with a score of zero, and positive and negative scores around the zero score, has also been suggested (http://v2020eresource.org/content/files/guideline_kap_Jan_mar04.pdf)

Scoring the most positive response option as 10, the least positive as zero, and the middle as 5 has also been recommended (https://www.cqc.org.uk/sites/default/files/20151125_nhspatientsurveys_scoring_methodology.pdf)

Comment: How was the practice level dichotomized? 

Response: Practice level was dichotomized as good and fair or poor practice as shown in Figure 3A & B. The phrase has been added to the “statistical analysis” section.

Results

Comment: A proportion of 43% cannot be regarded as majority. Use highest proportion instead. Correct in other areas. Majority is recorded for proportions above 50%. 

Response: The suggestion has been effected.

Comments:

Table 1

Second column should be frequency (n=156)

Remove the row total (Do the same for other tables).

Figures 3A and 3B should be removed. Then for table 5, remove the crude odds ratios and replace with proportions that had good and poor practice. The p value for the bivariate analysis should be included. For the multivariate analysis, present the adjusted odds ratio and 95% confidence interval. The p value on multivariate analysis may be included.

Remove the constants in table 5

All the variables had both crude and adjusted odds ratios. What qualified the variables for inclusion in the multivariate logistic regression model? If all the variables merited inclusion as seen in the result, what is the rationale for that?

Considering the small sample size as seen in this study, why should duration of practice in years have 5 categories? This remarkably affected the adjusted odds ratio and should be reviewed. Review the number of categories for other independent variables.

Response: The tables and figures are okay as presented and the style reflects the concept of the study. “n” connotes frequency and is the same as “f”. Some may chose to use “f”, others use “n”. Proportions are presented in figure 2 and must not be presented in table 5.

Please note that there are several criteria and methods for inclusion of variables in regression models and different scholars use different methods. The number of independent variables was modest so all were included in the model to minimize bias.

Discussion

Comment: Include Nigeria after Ogun state for better understanding 

Response: The suggestion has been effected.

Comment: Discuss the predictors of good practice of the National guideline 

Response: The investigator could not identify any similar studies that reported predictors of good practice for comparison and discussion. 

Conclusions

Comments: 

This should be in a single paragraph 

Use predictor of good practice instead of strong predictor

Response: The suggestions have been effected.

Reviewer # 3 

Title

Comment: The title does not match with the study objective. The title reflects a study which is purely descriptive, however, the objective of the study is to also assess factors associated with practices regarding the Malaria guideline. Kindly recast the title to reflect that the study assessed and reports about predictors. 

Response: The title is okay as stated. There is nothing in the title that suggests a purely descriptive study. The basic design is included in the title, viz: “a cross-sectional survey”, which could be descriptive or analytical. Some titles of similar studies in the literature includes “predictors” or “associated factors” and others do not. 

Introduction

Comment: Line 66: ….the risk of drug resistance. It should be “…………risk for development of drug resistance” 

Response: The phrase is correct. If risk is the probability that an event will occur, then the phrase is okay.

Comment: Line 87-88: The aim of this study was to evaluate the knowledge, attitude, and practice of the 2015 National Guidelines for Diagnosis and Treatment of Malaria, and associated factors, among medical doctors in Ebonyi state, Nigeria. Rephrase the title to match with the study aim. 

Response: The aim of the study must not be stated verbatim in the title and this is so in many articles in the literature. 

Methodology

Study design and setting

Comment: The study design should be cross-sectional analytical design. 

Response: I seem not to understand the comment. Perhaps the intended comment is that “cross-sectional” should come before “analytical” because “analytical” precedes “cross-sectional” as currently stated. That notwithstanding, the original statement has been modified in response to reviewer # 1. 

Study participants and data collection

Comment: The author states that they selected 156 medical doctors, what was the basis for selecting this number of participants, how sure were they that by engaging 156 medical doctors the study will be able to provide reliable answer to the questions it tries to answer? 

Response: As stated in the manuscript, convenience sampling of those that were available and disposable, was used to select the participants.

Results

Sociodemographic and background characteristics

Comment: When describing results, remove the word “of” after “the majority”, if there is a need of using the word “of” then it should be followed by “respondents” then number and percentage. Response: The sentence has been restated in line with the suggestion. 

Comment: Table 1 should be inserted after paragraph 1 that describes sociodemographic characteristics of respondents 

Response: The comment seems not to be clear. The table is located according to the guideline of the journal.

Comment: The way the total is indicated in table 1 (156) it as if 156 is the total of all the values for all the variables in the table. The best way is to indicate this total in the table title in brackets. This should be done for all the tables. 

Response: The way the “total” is located is also conventional. However, it has been moved to the title (but without bracket)

Comment: In table 1, on Place of practice as a variable should have two categories i.e. Secondary health facility and Federal university teaching hospital. This will make this table easier to comprehend. Then plot a bar chart for each of these categories to indicate the distribution of specific place of practices. 

Response: I do not think the table is difficult to understand. It is better to present the variable this way than plotting a separate graph for the subcategory.

Comment: To indicate less or equal use ≤ instead of <= 

Response: The suggestion has been effected.

Comment: The sum of the percentages of age categories gets to 100.1, please check Response: This was due to approximation (rounding-up) of individual values and is acceptable. However, adjustment has been made.

Knowledge of the 2015 National Guidelines for Diagnosis and Treatment of Malaria

Comment: Table 2 should be cited at the end of the paragraph 

Response: The table is located according to the guideline of the journal.

Comments: 

- Totals should be on every major category and should be indicated in brackets, i.e. “Knowledge items (156)”, “Among those that were aware there was a national guideline for diagnosis and treatment of malaria (138)”.

- “Among those that were aware there was a national guideline for diagnosis and treatment of malaria” this can be removed and remain with “Sources of information on the national malaria treatment guideline”

Response: Adjustments have been made as deemed appropriate.

Comment: Because items in table 2 are multiple responses items, putting a 100 on the percentage as total is potentially misleading because when you sum up individual percentages goes far beyond 100. 

Response: The total percentage has been removed.

Factors associated with good practice of the 2015 National Guidelines for Diagnosis and Treatment of Malaria

Comment: Why have you included all variable in your multivariable model? Usually, one would construct this using the variables with a predefined p-value from the univariable analysis. What was the criteria for inclusion of variables into the multivariable model? 

Response: Please note that there are several methods and criteria for inclusion of variables in regression models and different scholars use different methods. The number of independent variables was modest so all were included in the model to minimize bias.

Discussion

Comment: Authors repeats results description in the discussion section. Kindly revise the discussion section by discussing findings rather than rewriting the findings as it appears now. Response: The description of results was not repeated in the discussion. Rather, the relevant/suitable results were compared with those of other studies quantity by quantity. That is, the comparison had to be quantitative since quantitative studies were being compared. This is a better approach because readers can easily appreciate how the results/findings compare with those of other studies quantitatively. 

Conclusions

Comment: The authors recommend interventions that would improve attitude, it will be more meaningful to probably provide examples of such interventions. 

Response: Interventions such as training and re-training of the medical doctors on the national guidelines could improve their attitude. This has been added to the manuscript.

---

## [Decision Letter · Decision Letter 1]

2 Aug 2021

PONE-D-21-07204R1

Knowledge, attitude, and practice of the National Guidelines for Diagnosis and Treatment of Malaria among medical doctors in Ebonyi state, Nigeria: a cross-sectional survey

PLOS ONE

Dear Dr. Omale,

Thank you for submitting your manuscript to PLoS ONE. After careful consideration, we felt that your manuscript still requires substantial revision, following which it can possibly be reconsidered, thus governing the decision of a “major revision”. The reviewers felt that some important questions still need to be addressed. According to the reviewer #2, the authors did not incorporate in the revised MS most of the concerns raised previously.   A major concern raised by the reviewer #3 was related to the representativeness of the study sample to the general population of of medical doctors in Ebonyi.  At this time, we strongly suggest the authors to proper address all topics raised by the reviewers.  For your guidance, a copy of the reviewer’s comments was included below.  

We look forward to receiving your revised manuscript.

Kind regards,

Luzia Helena Carvalho, Ph.D.

Academic Editor

PLOS ONE

Reviewers' comments:

Reviewer's Responses to Questions

**Comments to the Author**

1. If the authors have adequately addressed your comments raised in a previous round of review and you feel that this manuscript is now acceptable for publication, you may indicate that here to bypass the “Comments to the Author” section, enter your conflict of interest statement in the “Confidential to Editor” section, and submit your "Accept" recommendation.

Reviewer #1: All comments have been addressed

Reviewer #2: (No Response)

Reviewer #3: (No Response)

2. Is the manuscript technically sound, and do the data support the conclusions?

Reviewer #1: Yes

Reviewer #2: Yes

Reviewer #3: No

3. Has the statistical analysis been performed appropriately and rigorously? 

Reviewer #1: Yes

Reviewer #2: Yes

Reviewer #3: No

4. Have the authors made all data underlying the findings in their manuscript fully available?

Reviewer #1: Yes

Reviewer #2: Yes

Reviewer #3: Yes

5. Is the manuscript presented in an intelligible fashion and written in standard English?

Reviewer #1: Yes

Reviewer #2: No

Reviewer #3: Yes

6. Review Comments to the Author

Reviewer #1: Thank you for your article improvement. It is now with a good standard, to the next step. Good luck.

Reviewer #2: Review comments

General comments

The Author responded to the comments by the Reviewers. This is good and commendable. However, most of the responses were not reflected in the revised manuscript. Most issues as raised by Reviewers are aimed towards improving the quality of the manuscript and not necessarily for the attention of Reviewers only. This was the observation from the comments of the Author. It should be noted that there are some issues that should be attended to before the manuscript could be accepted for publication.

In line with the above thoughts, the following points should be addressed;

1. The Author commented that convenience sampling was used in the selection of the respondents. This should be included in the abstract.

2. Author should state in the manuscript the basis for inclusion of variables in multivariate analysis after bivariate analysis. This is very important as it will enable readers have a good understanding of the research.

3. The author should take another look at the grading of the outcome variables. It was stated that <50% was designated as poor, 50-75% as fair and >75% as good. The clarification needed here is why 75% should be classified as fair.

4. Like I remarked in my previous comment, the sample size for the study is 156 which is relatively low. The 95% confidence interval for the (adjusted odds ratio) apart from indicating statistical significance also gives a precision of the estimate which is a reflection of the sample size.

Based on this observation, one wonders why the Author categorized the age groups into 4 groups, professional rank and duration of service into 5 groups each. However, the Author categorized primary place of practice into 2 groups.

It is good to take a look at the 95% confidence interval in the multivariate analysis for the age group, >50 years, it is 0.1-91.3. That for >20 years of service is 0.0003-3.3.

Author should endeavor to dichotomize all independent variables in the bivariate and multivariate analysis. There should be a good explanation for categorizing any independent variable into three groups.

Abstract

Methods

The author should include how the respondents were selected

Introduction

Line 79, page 4, correct malarian to malaria

Methods

Line 92, page 4, remove the

Line 109, change the word disposable

Reviewer #3: Review Comments on a Manuscript Titled: Knowledge, attitude, and practice of the National Guidelines for Diagnosis and Treatment of Malaria among medical doctors in Ebonyi state, Nigeria: a cross-sectional survey

For the title to be more effective, it must contain the main topics of the study, in this case, the word “predictors” for knowledge, attitude and practice key in this study and it must surface in the title in any form.

It is best to have the title and objective of the study marrying one another, I recommend the title be rephrased to match with the study aim.

The author states that they selected 156 medical doctors, what was the basis for selecting this number of participants? Why didn’t they just engage 100 medical doctors? How representative is the sample to the general population of medical doctors in Ebonyi state, much as the findings from 156 medical doctors is inferred to all the medical doctors in Ebonyi state?

7. PLOS authors have the option to publish the peer review history of their article (what does this mean?). If published, this will include your full peer review and any attached files.

Reviewer #1: **Yes: **Adilson José DePINA

Reviewer #2: **Yes: **EDMUND NDUDI OSSAI

Reviewer #3: No

---

## [Author Response · Author response to Decision Letter 1]

5 Aug 2021

RESPONSE TO REVIEWERS

Reviewer # 1 

Comment: Thank you for your article improvement. It is now with a good standard, to the next step. Good luck. 

Response: Best wishes in your every endeavor.

Reviewer # 2 

1. Comment: The Author commented that convenience sampling was used in the selection of the respondents. This should be included in the abstract. 

Response: The suggestion has been effected.

2. Comment: Author should state in the manuscript the basis for inclusion of variables in multivariate analysis after bivariate analysis. This is very important as it will enable readers have a good understanding of the research. 

Response: All the independent variables were simultaneously entered into the multivariate regression. The number of independent variables was modest so all were included in the model to minimize bias. This has been added to the “statistical analysis” section for more clarity.

3. Comment: The author should take another look at the grading of the outcome variables. It was stated that <50% was designated as poor, 50-75% as fair and >75% as good. The clarification needed here is why 75% should be classified as fair. 

Response: The author is not aware of any recommended cut-off for grading KAP of health workers. Even if “70%” or “60%” is used (instead of “75%”), the same question could still be posed as to: why 70%? Why 60? It should be noted that different authors have used different cut-offs and some categorized scores greater than 75% as good KAP or even 80% and above as good adherence. Examples:

1. Usman R. et al. Predictors of malaria Rapid Diagnostic Tests’ utilisation among healthcare workers in Zamfara State. PLoS One. 2018;13:12.

2. Kotepui KU et al. Knowledge, Attitude, and Practice Related to Malaria Diagnosis among Healthcare Workers in Hospitals: A Cross-Sectional Survey. Journal of Tropical Medicine. Volume 2019.

3. Na’uzo AM et al. Adherence to malaria rapid diagnostic test result among healthcare workers in Sokoto metropolis, Nigeria. Malaria J. 2020;19:2

Moreover malaria is a common illness in Nigeria, and the respondents in this study were medical doctors who are highly skilled medical practitioners that are expected to uphold optimal patient management, and this is the aim of having diagnosis and treatment guidelines. The scoring and grading approach used was consistent with the concept of the study as described under the “introduction section” of the manuscript.

4. Comment: Like I remarked in my previous comment, the sample size for the study is 156 which is relatively low. The 95% confidence interval for the (adjusted odds ratio) apart from indicating statistical significance also gives a precision of the estimate which is a reflection of the sample size.

Based on this observation, one wonders why the Author categorized the age groups into 4 groups, professional rank and duration of service into 5 groups each. However, the Author categorized primary place of practice into 2 groups.

It is good to take a look at the 95% confidence interval in the multivariate analysis for the age group, >50 years, it is 0.1-91.3. That for >20 years of service is 0.0003-3.3.

Author should endeavor to dichotomize all independent variables in the bivariate and multivariate analysis. There should be a good explanation for categorizing any independent variable into three groups.

Response: Even though the width of the confidence interval for a number of categories (>50 years of age, >20 years duration of practice) are very wide, the width of the confidence interval for most of the other categories are modest considering the relatively small sample size and dichotomizing all the variables will result in great loss of information and may also result in loss of power in detecting any difference in the overall effect. Comparing two levels of care (secondary and tertiary) makes great sense and does not indicate that other variables must be categorized into two. For instance, there is no way the professional rank will be grouped into two that it will make any great theoretical sense. There is no recommendation that all independent variables should have equal or two categories before running regression analysis. However, because of the concern over the wide confidence intervals, the “>50 years of age” and “>20 years duration of practice” categories have been merged with the preceding categories respectively to produce 3 categories of age group and 4 categories of duration of practice . It is important to note that exploratory categorization of age group and duration of practice into only two groups did not produce much changes in the width of the confidence interval compared to the current 3 categories of age group and 4 categories of duration of practice.

Comment: Introduction: Line 79, page 4, correct malarian to malaria 

Response: The typo has been corrected

Comments: Methods

Line 92, page 4, remove the

Line 109, change the word disposable

Response: The typo has been corrected and “disposable” has been replaced.

Reviewer # 3 

Comments: For the title to be more effective, it must contain the main topics of the study, in this case, the word “predictors” for knowledge, attitude and practice key in this study and it must surface in the title in any form. It is best to have the title and objective of the study marrying one another, I recommend the title be rephrased to match with the study aim. 

Response: There are no reporting guidelines that recommend all components of the study aims and objectives must be included in the title of the study. The concept of the study did not capture “predictors” as the main topic. The purpose and basic design of the study has been captured in the title like those of many other similar articles/studies, examples:

1. Riley C et al. Knowledge and Adherence to the National Guidelines for Malaria Case Management in Pregnancy among Healthcare Providers and Drug Outlet Dispensers in Rural, Western Kenya. PLoS One. 2016;11(1): e01

2. Blanco M et al. Knowledge and practices regarding malaria and the National Treatment Guidelines among public health workers in Equatorial Guinea. Malar J. 2021;20:21

3. Na’uzo AM et al. Adherence to malaria rapid diagnostic test result among healthcare workers in Sokoto metropolis, Nigeria. Malaria J. 2020;19:2

4. Riley C et al. Knowledge and Adherence to the National Guidelines for Malaria Diagnosis in Pregnancy among Health-Care Providers and Drug-Outlet Dispensers in Rural Western Kenya. Am. J. Trop. Med. Hyg. 2018;98(5):1367–1373

5. Okoro et al. The Cross-Sectional Evaluation of the Use of Artemisinin-Based Combination Therapy for Treatment of Malaria Infection at a Tertiary Hospital in Nigeria. Journal of Tropical Medicine. Volume 2018. 

6. Mokuolu OA et al. Provider and patient perceptions of malaria rapid diagnostic test use in Nigeria: a cross‑sectional evaluation Malar J. 2018;17:200

7. Akinyode AO et al. Practice of antimalarial prescription to patients with negative rapid test results and associated factors among health workers in Oyo State, Nigeria. Pan African Medical Journal. 2018; 30:229

8. Bamiselu OF et al. Adherence to malaria diagnosis and treatment guidelines among healthcare workers in Ogun State, Nigeria BMC Public Health. 2016;16:828

Comment: The author states that they selected 156 medical doctors, what was the basis for selecting this number of participants? Why didn’t they just engage 100 medical doctors? How representative is the sample to the general population of medical doctors in Ebonyi state, much as the findings from 156 medical doctors is inferred to all the medical doctors in Ebonyi state? Response: It is already stated in the manuscript that convenience sampling was employed in selection of participants. The 156 medical doctors were the ones that were available (during a visit to their respective places of work) and consented to participate in the study. If more or less were available and consented, more or less would have participated. Moreover, 156 is a reasonable percentage of the estimated 670 medical doctors in the Ebonyi state (according to the state chapter of the Nigerian Medical Association)

---

## [Decision Letter · Decision Letter 2]

6 Sep 2021

Knowledge, attitude, and practice of the National Guidelines for Diagnosis and Treatment of Malaria among medical doctors in Ebonyi state, Nigeria: a cross-sectional survey

PONE-D-21-07204R2

Dear Dr. Omale,

We’re pleased to inform you that your manuscript has been judged scientifically suitable for publication and will be formally accepted for publication once it meets all outstanding technical requirements.

Kind regards,

Luzia Helena Carvalho, Ph.D.

Academic Editor

PLOS ONE

Additional Editor Comments (optional):

Reviewers' comments:

Reviewer's Responses to Questions

**Comments to the Author**

1. If the authors have adequately addressed your comments raised in a previous round of review and you feel that this manuscript is now acceptable for publication, you may indicate that here to bypass the “Comments to the Author” section, enter your conflict of interest statement in the “Confidential to Editor” section, and submit your "Accept" recommendation.

Reviewer #1: All comments have been addressed

Reviewer #2: All comments have been addressed

2. Is the manuscript technically sound, and do the data support the conclusions?

Reviewer #1: Yes

Reviewer #2: Yes

3. Has the statistical analysis been performed appropriately and rigorously? 

Reviewer #1: Yes

Reviewer #2: Yes

4. Have the authors made all data underlying the findings in their manuscript fully available?

Reviewer #1: Yes

Reviewer #2: Yes

5. Is the manuscript presented in an intelligible fashion and written in standard English?

Reviewer #1: Yes

Reviewer #2: Yes

6. Review Comments to the Author

Reviewer #1: The authors have adequately addressed your comments raised in a previous round of review and I feel that this manuscript is now acceptable for publication

Reviewer #2: (No Response)

7. PLOS authors have the option to publish the peer review history of their article (what does this mean?). If published, this will include your full peer review and any attached files.

Reviewer #1: **Yes: **Adilson DePINA

Reviewer #2: **Yes: **Edmund Ndudi Ossai